# The Digital Calibration Certificate (DCC) for an End-to-End Digital Quality Infrastructure for Industry 4.0

Siegfried Hackel [1,*], Shanna Schönhals [1], Lutz Doering [1], Thomas Engel [2] and Reinhard Baumfalk [3]

1   Physikalisch-Technische Bundesanstalt (PTB), 38116 Braunschweig, Germany
2   Siemens Aktiengesellschaft, 81739 München, Germany
3   Sartorius Lab Instruments GmbH & Co. KG, 37079 Göttingen, Germany
*   Correspondence: siegfried.hackel@ptb.de; Tel.: +49-531-592-1017

**Abstract:** This article depicts the role of the Digital Calibration Certificate (DCC) for an end-to-end digital quality infrastructure and as the basis for developments that are designated by the keyword "Industry 4.0". Furthermore, it describes the impact the DCC has on increasing productivity in the manufacturing of products and in global trade. The DCC project is international in its scope. Calibration certificates document the measurement capability of a measurement system. They do this independently and by providing traceability to measurement standards. Therefore, they do not only play an important role in the world of metrology, but they also make it possible for manufacturing and commercial enterprises to exchange measurement values reliably and correctly at the national and at the international level. Thus, a DCC concept is urgently needed for the end-to-end digitalization of industry for the era of Industry 4.0 and for Medicine 4.0. A DCC brings about important advantages for issuers and for users. The DCC leads to the stringent, end-to-end, traceable and process-oriented organization of manufacturing and trading. Digitalization is thus a key factor in the field of calibration as it enables significant improvements in product and process quality. The reason for this is that the transmission of errors will be prevented, and consequently, costs will be saved as the time needed for distributing and disseminating the DCCs and the respective calibration objects will be reduced. Furthermore, it will no longer be necessary for the test equipment administration staff to update the data manually, which is a time-consuming, tedious and error-prone process.

**Keywords:** D-SI; DCC; digital signature; calibration; Industry 4.0

## 1. Introduction

The analogue calibration certificate is currently still used and issued in paper form or as a closed PDF document. These closed documents hardly contribute to improving the production process of a company. The reason for this is that using analogue calibration data from a calibration in subsequent processes is very time-consuming and prone to errors as these data have to be converted once again into the digital formats of the specific manufacturing plant. The DCC [1] overcomes this disadvantage of its analogue counterpart. The objective was to create an internationally recognized DCC format which acts as an interface (exchange format) in the whole field of metrology, and especially in the field of machine-to-machine communication. On the basis of the DCC, we will be able to develop further exchange formats in the future. These might be in the field of legal metrology, for digital type examination certificates, for the **D**igital **T**win (DT) or for developments in many other fields—for example, the **D**igital **C**alibration **R**equest (DCR) format or, in the field of automated accident notification, in the format of the "**I**nternational **S**tandard **A**ccident **N**umber (ISAN): Linking data in accidents and emergencies." The DCC serves for the electronic storage, the authenticable and—if necessary—the encrypted and signed dissemination and the uniform interpretation of the calibration results. Due to the DCC schema to be applied, the DCC is thus both machine-readable and machine-interpretable



when using Good Practice (GP) DCCs. Further information on machine interpretability can be found in [2]. The DCC has been developed and agreed upon in a broad community and is constantly being developed further in order to achieve worldwide acceptance. The target groups are all those bodies, authorities and companies worldwide which require proof of the metrological traceability of their measurement results and use these results in modern manufacturing processes in the field of IIoT/Industry 4.0. These bodies, authorities and companies include the metrology institutes, designated institutes, national calibration centres, calibration laboratories and the many companies in industry that require traceable measurement results for their quality management systems.

According to the International Vocabulary of Metrology (VIM), the term "calibration" is defined as follows [3]:

"Operation that, under specified conditions, in a first step, establishes a relation between the quantity values with measurement uncertainties provided by measurement standards and corresponding indications with associated measurement uncertainties and, in a second step, uses this information to establish a relation for obtaining a measurement result from an indication" [4], p. 3.

The ISO IEC 17025 standard [5] describes the general requirements that are placed on qualified calibrations. Section 7.8 of this standard deals with "reporting on results." Although digitalization is increasing worldwide, calibration certificates are still mainly issued in analogue form, meaning that they are generated in paper form. This does not only lead to numerous problems, but it also prevents, in many fields, the complete digitalization of the value chain, not least due to the fact that analogue calibration certificates are often designed differently, even if the calibrations are the same. A uniform design for certificates for calibrations which are basically equal in nature is shown in a Good Practice (GP) DCC. Work on the GP-DCC is currently in process in the most diverse fields (temperature, humidity, ambient pressure, mass, weighing instruments, force, torque...). By means of the GP-DCC, the results are becoming machine-interpretable (as explained above) as, thanks to the GP-DCC, it is exactly known what has been stored, where it has been stored and in which way it has been stored.

Calibrations—and thus also calibration certificates—play an important role in many branches of industry (e.g., in the automotive industry and the pharmaceutical industry). A pharmaceutical company may require several hundreds of thousands of calibrations certificates every year. It is easy to imagine which efforts currently have to be made to archive analogue calibration certificates (either exclusively in paper form or also in other forms). In this publication, the DCC concept is presented. It will solve the abovementioned problems and will bring about many additional advantages for companies and other users. Furthermore, it will allow machines to communicate with each other (M2M). All of these aspects are described in more detail below.

## 2. The Role of Calibration and Calibration Certificates

### 2.1. Quality Infrastructure

The quality infrastructure of a country is of fundamental importance for the services that are provided to the public as well as for consumer protection. This was published for the first time in 2007 in a report of the World Bank [6]. Meanwhile, various contributions have been issued which were developed on the basis of this report (see also [7–10]). Figure 1 shows the general set-up of an arbitrary national infrastructure, and it also shows digitalization by means of DCC. From this figure, it becomes obvious what an important role calibration certificates play.

Calibration certificates appear several times in Figure 1. It should be stated that besides the "normal" digitalization effect which normally occurs, for example, in the field of eGovernment, an exorbitant increase in quality can be observed, which is due to the fact that the calibration information is more precise and can be used in an automated way. The data are transferred in M2M communication without any transmission errors. The step in which data are transferred manually according to the four-eyes principle and

which so far has been necessary, for example, in the pharmaceutical industry, can then be omitted. Thereby, the time expenditure can be substantially reduced, and errors during data transmission will thus be ruled out.

# National Quality Infrastructure

**Figure 1.** Presentation of the national quality infrastructure (based on [6]).

After the data have been transferred, the reject rate in manufacturing can immediately be reduced and products can be manufactured in a more resource-saving and more sustainable way. Manufacturing is increasingly helping to save many raw materials, as well as time and energy, and thus often reduces $CO_2$ emissions to a high extent. This, in turn, will contribute essentially to environmental protection and to achieving the global UN sustainable development goals not only in the industrial countries, but also in developing countries and in countries in transition. The latter can often even skip an innovation cycle (which would normally still be based on a paper-based calibration system) and thus participate more quickly in an international economic system. In addition, they could contribute effectively to the conservation of nature especially in their own countries.

The topic of "data as economic goods" has become particularly important in the value chain of industrial companies and will become more and more important in the future due to the advancing digital transformation. In this regard, the quality of the data which have been gained, among others, via sensors and actuators and the verifiability of the statements and conclusions derived from this data are of central importance especially for industry. It will only be possible to use data successfully as economic goods if the quality of the data is reliably verified and safeguarded.

For more than 125 years, maintaining the high level of quality of German products has been one of the fundamental pillars of the quality label "Made in Germany" which is recognized worldwide today. One of the fundamental challenges of the national quality infrastructure (QI) is to transfer this quality label to the digitalized world and to establish it there. This includes, in particular, the development of safe and robust calibrated measurement systems to be able to ensure data quality and the trust people have in the data. Metrology plays a decisive role in gaining the trust people have in measurements and in

ensuring the quality of the measurement data and measurement results. Figure 1 shows how the different elements of QI interact with each other.

The quality infrastructure of a country consists of seven elements. The **N**ational **M**etrology **I**nstitute (NMI) disseminates the SI units to the national standardization institute. The standardization institute ensures that the level of the enterprises (see the level "enterprises" in the diagram above) is provided with norms and standards. At the same time, it determines the norms and standards for the national accreditation body. The accreditation body has the task of accrediting calibration and testing laboratories, inspection bodies and certification bodies and of supervising the accreditation. In this way, the accreditation bodies are able to supply the level of the enterprises with verified expertise.

The second task of the national metrology institutes is to provide the accredited calibration laboratories with traceable systems. In this way, it is ensured that the measurement standards of the accredited calibration laboratories are linked up with the national measurement standards. The accredited calibration laboratories will then ensure the traceability of the measurement standards of the test laboratories, of the inspection bodies and of the other calibration laboratories. Calibration thus has a special position in the national quality infrastructure (see the red arrows in Figure 1).

So far, the certification bodies have not been particularly active in the field of digitalization. However, it has turned out that especially in the field of software development, considerable damage has occurred. This is due to the fact that different systems of units are used. The software certification bodies will therefore play a particularly important role in this field.

DCCs have not yet been the subject of considerations. However, using DCCs, and thanks to the fact that the error-free transmission of data becomes possible using DCCs, a high increase in quality will be achieved as the transmission problems will be eliminated.

In addition, there is the advantage that in a DCC, larger amounts of data can be transmitted, and that the calibration information can be integrated in a **D**igital **T**win (DT) in an automated and secure way.

### 2.2. Task of the Calibration

The calibration of measurement instruments is the main pillar of measurements which are comparable with each other and whose contents are correct. It is also one of the main pillars of the distributed manufacturing processes that are common in industry today. Without calibration, it is impossible to make a sound statement on the quality of a manufactured product. The calibration is based on the measurement standards provided by the NMIs and the **D**esignated **I**nstitutes (DIs). These measurement standards are embedded in the SI system of units [11]. Figure 2 illustrates this relationship. In many calibration fields, the factors indicated in the figure are clearly higher. It can be assumed that in Germany alone, several millions of calibrations are carried out every year. Via large-scale comparison measurements (called "interlaboratory comparisons"), as well as via audits and other procedures, the NMIs and the DIs ensure the provided measurement standards agree within the scope of the conventional and/or stated measurement uncertainties. In addition to this, the NMIs and the DIs of different countries stay in close contact with each other in order to ensure there are harmonized measurement standards worldwide. The calibration pyramid shows that the measurement uncertainty increases from the top to the bottom.

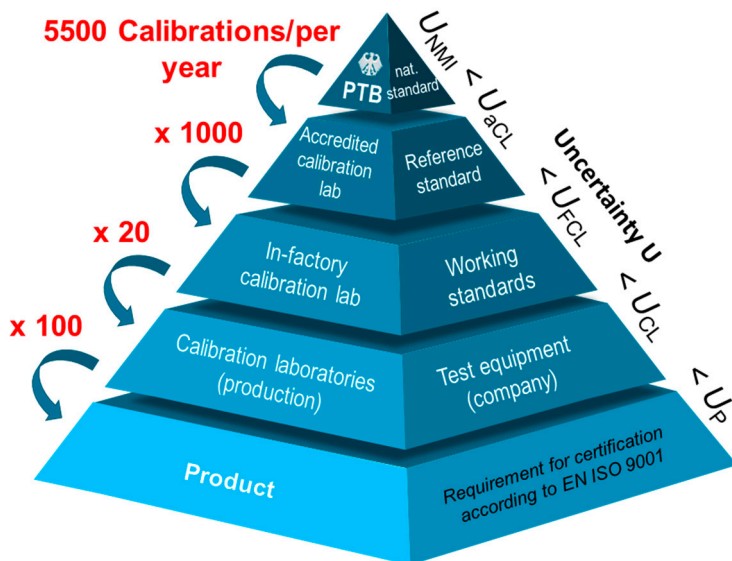

**Figure 2.** Representation of the calibration pyramid of the individual national metrology institutes/designated institutes and their integration in international interlaboratory comparisons with other NMIs/DIs.

## 3. Research for Practical Applications

Both industry and the economy are waiting for—and urgently require—a successful and lawful digital transformation so that data, information and certificates can securely and robustly be transmitted throughout the QI processes. Such a digital transformation is the precondition for both industry and the economy to be able to keep their promise of quality also efficiently and effectively in a digitalized world. Since August 2020, a project consortium has been established in Germany which is known as GEMIMEG-II [12] and is funded by the BMWK (Federal Ministry for Economic Affairs and Climate Action). It consists of 12 partners as well as of further institutions (the latter in a consultative capacity). The partners are contributing research achievements in the field of DCC [13,14] and are setting up a digital calibration infrastructure. The overriding objective and the central challenge of the GEMIMEG-II project is to bring the framework requirements of QI and the practical demands of industry together in a joint, holistic approach which is applicable in practice.

The focus of this project is on metrology (calibration, referencing, measurement, i.e., on trustable information on the quality of measuring instruments, sensor networks, digital twin and data analysis methods). The overall objective is to make information for the realization of reliable, connected measuring systems available in a secure, lawful and legally compliant end-to-end way. The results of the research fields are implemented in four so-called "RealBeds" (i.e., demonstrators which are geared to different fields of application), and these demonstrators thus take on the status of a prototype. The four RealBeds are connected calibration facilities, Industry 4.0 applications, the pharmaceutical/process industry and autonomous driving. The legal questions of the research fields and of the RealBeds will be investigated in addition in a legal simulation study.

In these investigations, two aspects are important: (1) In how far is reliable and trustworthy information on measuring instruments and measurement data digitally available, and (2) how can a secure and robust orchestration of the measuring systems be achieved? The investigations also include the development of a digital metrology system in the sense of an end-to-end, digitalized, traceable and legally secure measuring and calibration chain for complex sensor networks. Thereby, the sensors—which, so far, have been considered individually—will increasingly be connected with each other in a network by the IIoT (**I**ndustrial **I**nternet **of T**hings) in Industry 4.0. A basic set-up of a generic, massive, connected sensor system is shown in Figure 3. The processing of the sensor data follows the

fog/edge computing approach. This allows computing functions and memory assignments to take place on suitable nodes of a network, but completely decentralized solutions for use in mobile systems are also possible by means of identical stack and software technologies.

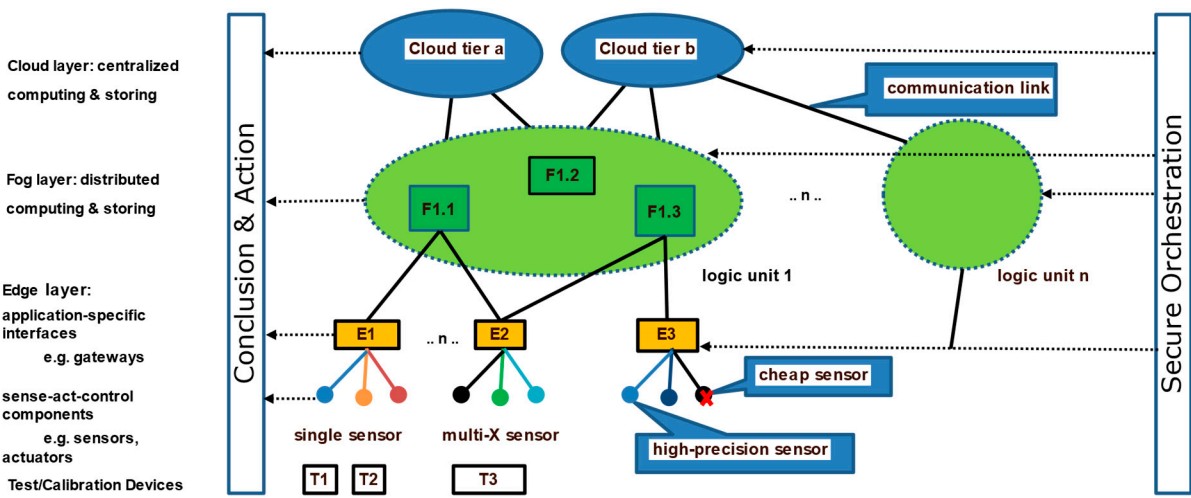

**Figure 3.** Architecture for a massive sensor network based on the OpenFog reference architecture [15,16].

This technical and automated interconnection in a network significantly increases the requirement to determine the quality of data and their availability if, on the one hand, the integrity of the citizens and of the environment and, on the other hand, the economic success of the companies is to be ensured. It can thus be expected that the need for developing and setting up sensor networks as holistically calibrated, digital measuring systems will continuously increase.

At this point, special requirements will not only arise for the quality of data, but also for the communication infrastructure and for the management of data sources and data sinks. Figure 4 shows a schematic overview of the GEMIMEG-II concept.

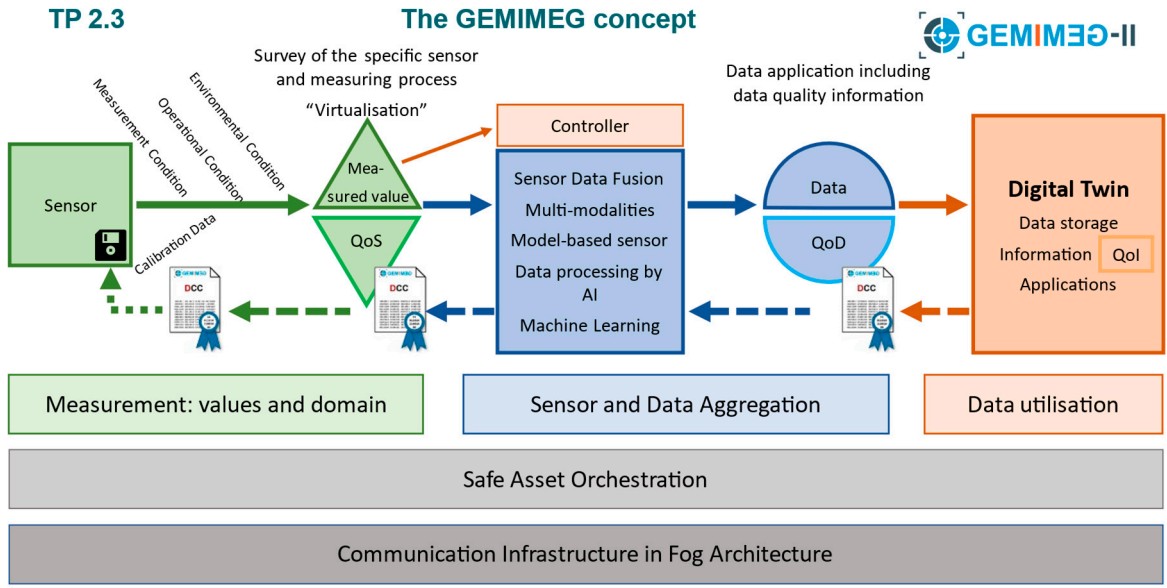

**Figure 4.** GEMIMEG concept in a schematic overview.

This concept extends from the sensor measurement values, including calibration and the characteristic numbers QoS (QoS: **Q**uality **o**f **S**ensing), via sensor fusion and data aggregation with the characteristic numbers QoD (**Q**uality **o**f **D**ata) to data application

(e.g., in the digital twin) with the characteristic numbers QoI (**Q**uality **o**f **I**nformation). The quality characteristic numbers can also be described as QoX, where the X stands for **S**ensing, **D**ata, or **I**nformation.

That step of the measurement process which contains the sensor measurement values and the calibration of the sensors is supposed to abstract the measurement values (by exploiting the domain knowledge during the measurement process) and to decouple them from the specific characteristics of the actually used sensor. The qualified measurement values and the quality characteristic numbers which have been obtained in this way in the subsequent step (sensor and data aggregation) can be processed without the concrete measurement system being known any further. The measurement values are thus decoupled from the corresponding measurement modalities in the best possible way and associated QoS trust measures have been determined. After these input quantities have been processed during the sensor and data aggregation step, new data are generated as output quantities—after data fusion, or via software-based sensors, or via data analysis using neural networks, or via artificial intelligence—which, in turn, can then have the quality characteristic numbers QoD. This step can be available several times. It can be arranged in parallel (for different measurement values) and/or in a cascaded way (multi-step processing). The data with QoD can be used in different digital twins which, in turn, can access the data with QoD from the multi-step sensor and data aggregation.

If pieces of information are then derived from these data (in an application such as a digital twin), this information can also be supplemented by means of quality characteristic numbers for the QoI information—where the QoI describes, for example, a level of trust or the reliability of this information.

The DCCs, which have already been addressed several times in this article, bring about several conceptual advantages for the project. Avoiding changes between different media—and thus ruling out transmission errors—is an obvious advantage. The calibration results are immediately available for further processing. It is thus, for example, possible to extract different measuring uncertainties automatically in the areas that are important for the manufacturing process.

## 4. The DCC Concept

In its digital form, the DCC is defined via an XSD schema [13]. For the information obtained in a calibration, the schema contains clearly defined fields for entering mandatory specifications, as well as open text fields in which the calibration laboratory can enter further information. In this way, the entire information content obtained during a calibration will be included in the DCC and transmitted.

The Digital System of Units (D-SI [17,18]) is used to express the units for the measurement values. It supports both the seven base units and the measurement units derived from these—including decimal multiples or dividers, among others—as well as the disseminated imperial units. Even unusual systems of units are possible as long as they have been clearly defined. Actually, this is necessary today in order to ensure, on the one hand, international acceptance, and to be able, on the other hand, to cope without difficulties with all the applications that exist today.

Figure 5 shows an initial rough generic view of the way in which the DCC is embedded in the existing environment of norms, standards, terms and technical regulations.

In addition to the DCC, Figure 5 also shows the **D**igital **C**alibration **R**equest (DCR). This is a standardized digital document by means of which a desired calibration can be requested and specified. Furthermore, Figure 5 shows a document named the **D**igital **C**alibration **A**nswer (DCA). In the DCA, the calibration laboratory can enter further information on the calibration which—for formal reasons (e.g., according to the ISO IEC 17025 standard [5])—is not supposed to appear in an official calibration document. The aim is that all documents should be based on a joint structured document schema DX which, in turn, must take the norms, standards, nomenclatures and technical regulations into account which are shown in Figure 5 below the level "Digital Document Schema DX." The purpose

of this is to ensure that the calibration is documented digitally in a way that is legally valid and in compliance with the norms and standards.

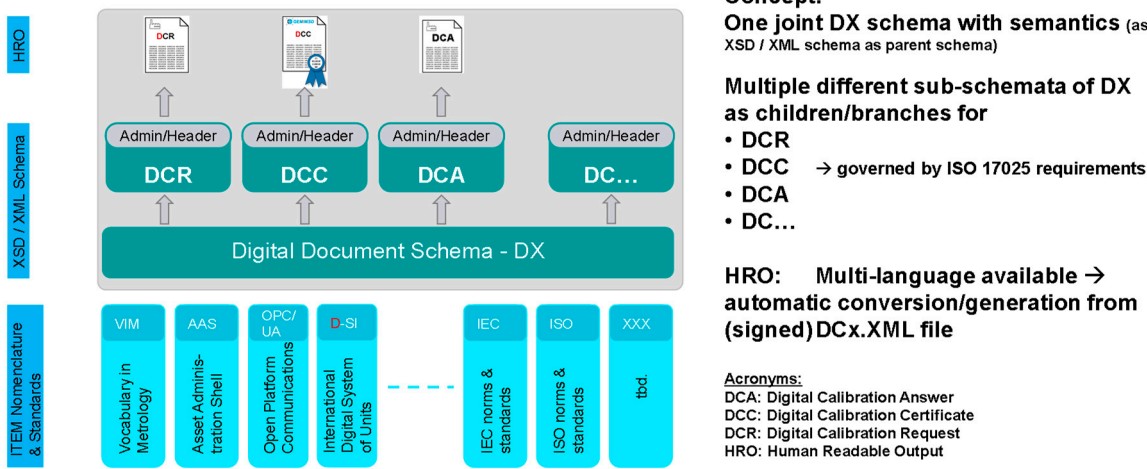

**Figure 5.** Generic view of the digital calibration document "ecosystem".

### 4.1. Structure of the DCC Files

Technically, the DCC is a text document which has an XML structure. XML documents are stored as plain text files, based on the Unicode character set (UTF-8 format [19]), and are thus suitable for long-term storage. The files can be opened and processed with an arbitrary text editor. Version 1.0, which is used for DCC, is widespread and has been standardized for more than 20 years now, which is why XML is ideally suited for the use of such important documents as calibration reports. The calibration reports can be ported from XML to other data formats (e.g., JSON).

More information on the structure of a DCC can be found on the DCC homepage [20].

### 4.2. Prologue including a Processing Instruction

Every XML document must start with a single-spaced prologue. The prologue is an instruction which has the form (<?xml...?>) and is named XML **P**rocessing **I**nstruction (PI). The prologue gives the reading program instructions on what must be observed during reading. Apart from the version (which can be indicated via the "version" attribute), it is also the agreed character set which can be stored in the document file. The characters in documents are encoded according to various ISO standards. The ISO 8859 standard [21] specifies the characters of different languages in ten subdivisions. Further characters are defined in ISO standard IEC 10646 [22]. DCC uses the character set UTF-8. This character set can map all official languages in the world.

One of the main advantages of XML is that it can be read and interpreted by machines. In addition, it is also readable by humans. A decisive factor is furthermore that XML is a data format that is suitable for long-term storage. This is of special importance as it must still be possible to read the files in several decades. Further information on data formats that are suitable for long-term storage can be found, for example, in [23,24].

### 4.3. Cryptographic Signatures in the DCC

In order to ensure the authenticity and the originality of each calibration certificate, the XML document can be provided by the issuing body (the calibration body) with a corresponding hash value and can be signed digitally. According to ISO standard 17025, adding a signature is **not mandatory**. As a measure to create trust between the calibration institute und the customer, however, adding a signature is recommended. As the signature

can be verified at any time, the recipients of a DCC are in a position to verify the authenticity of the content of a DCC as well as the issuer of a DCC themselves. For this purpose, public software tools are available free of cost [25]. The cryptographic procedures can be applied to XML data structures in a robust and probative way [23,26]. Further information on XML is to be found in the literature, e.g., in [27]. The VDI/VDE has already developed a very broad approach for data exchange [28] and—although this is a national standard for Germany—it is already used in various other countries. In this approach, the workflow in industry, in which the DCC can be integrated without any problems, is taken into account. On the one hand, the integrity and authenticity of a DCC must be ensured. On the other hand, however, electronically stored data can easily be changed and/or copied as often as desired. The use of cryptographic security procedures for the DCC is therefore highly recommended. Good summaries on the topic of cryptography can be found in [29,30].

Unfortunately, this report cannot describe this concept in more detail.

## 5. The DCC within the Scope of the Manufacturing Process

In this section, we, first of all, show the processes which take place within the scope of discrete manufacturing. In Figure 6, the logic relationship between the "parts supplier," the "machine" and the "factory" is shown in the form of models. All models include, in particular, commissioning and production processes.

**Figure 6.** Hierarchy of the different process models in the context of a factory, similar to [31] and adapted.

For these processes, calibrated measurement components are required. However, different pieces of information on the calibration are currently only available as calibration certificates in paper form (Figure 7).

When discrete or continuous manufacturing processes are automated, this offers considerable potential for using a DCC. Whereas in conventional automation with static hierarchies, it is sufficient to manage the calibration data of process sensors analogously and/or decentrally (Figure 8). Due to the fact that there are only a few clearly defined input points for this information, the situation in an Industry 4.0 architecture is different.

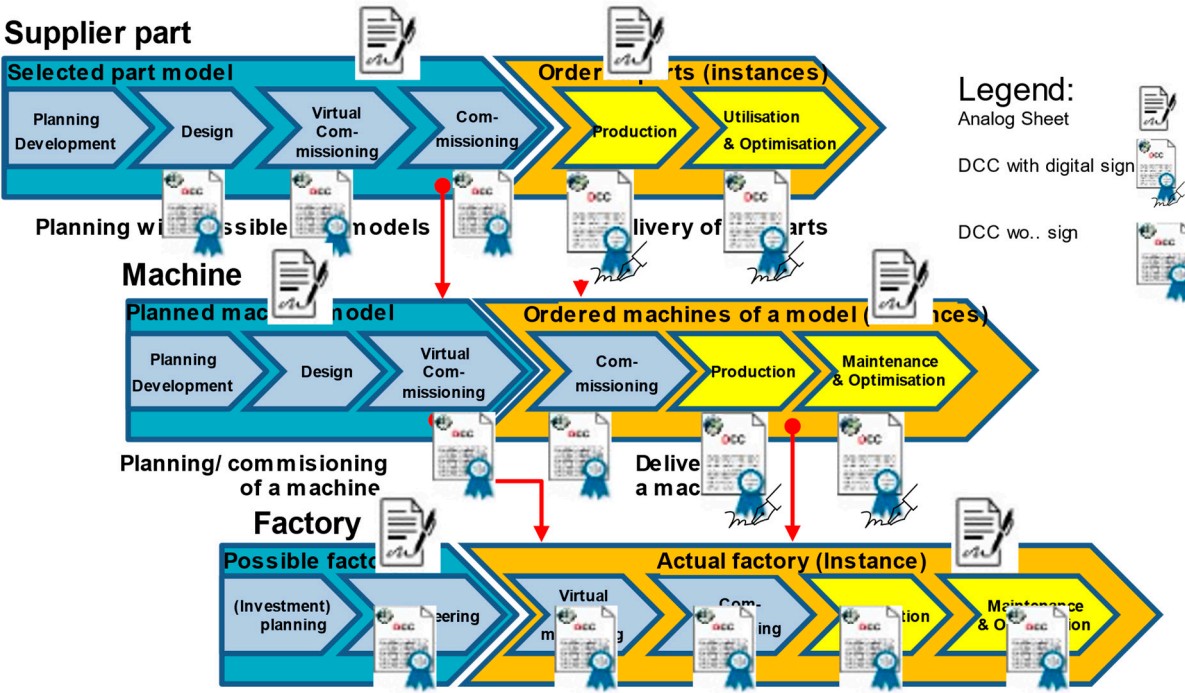

**Figure 7.** Comparison of how far the calibration information is available in an analogue calibration certificate (in the diagram in Figure 7, this is each time shown above the respective model) and in a DCC.

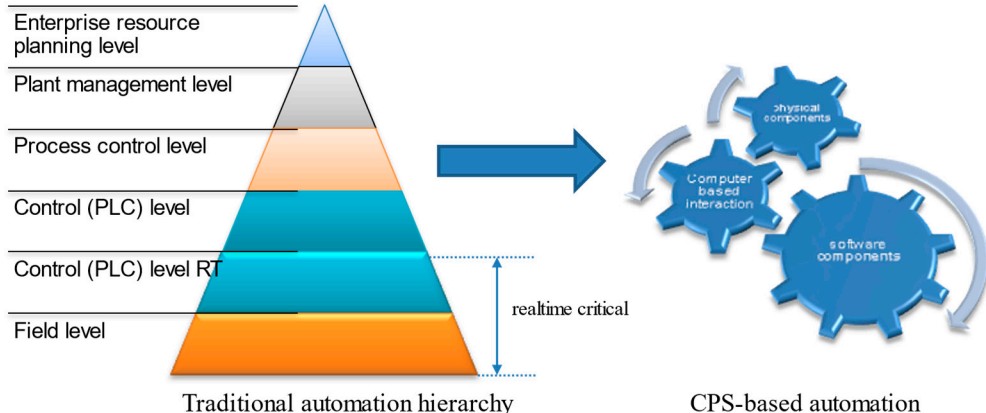

**Figure 8.** Conventional automation architecture as compared to Industry 4.0 or IIoT structures based on a Cyber Physical System (CPS) based on [32].

CPS-based architectures [33] react adaptively and continuously if it becomes necessary to change manufacturing processes, and this is exactly what is stipulated for Industry 4.0 or IIoT networks. The consequence of this is as follows: The process sensors, which are also used in automation, must not be able any longer to only make the generated measurement values but also—as an independent element—the sensor information (such as calibration certificates) available to other process components independently and in changing architectures. The massive sensor networks, which have already been presented above, might be an option for this and—together with the DCC—fulfil the requirements which have been discussed here.

Figures 7 and 8 clearly show that the calibration information that is contained in a DCC is not only available in the immediate manufacturing processes in which the calibrated measuring components are used. As the DCCs can be seamlessly integrated into the digital infrastructure of a factory, the calibration information can also be used in other processes

such as in enterprise planning, commissioning or quality management. The VDI/VDE has already developed a very broad approach for data exchange [28] and—although this is a national standard for Germany—it is already used in various other countries. In this approach, the workflow in industry, in which the DCC can be integrated without any problems, is taken into account.

Within the scope of the GEMIMEG-II project, we are currently working on a software library in the programming language Python (PyDCC). The aim of PyDCC is to facilitate access to the contents of the DCC and their processing so that the advantages of the DCC can be prepared for digital manufacturing in a simple and easy way. As soon as the GEMIMEG-II project has been concluded, this software package, which will be an open-source software, will become generally available. Figure 9 shows the current development and planning status of this software project as well as the focal areas of the future development tasks.

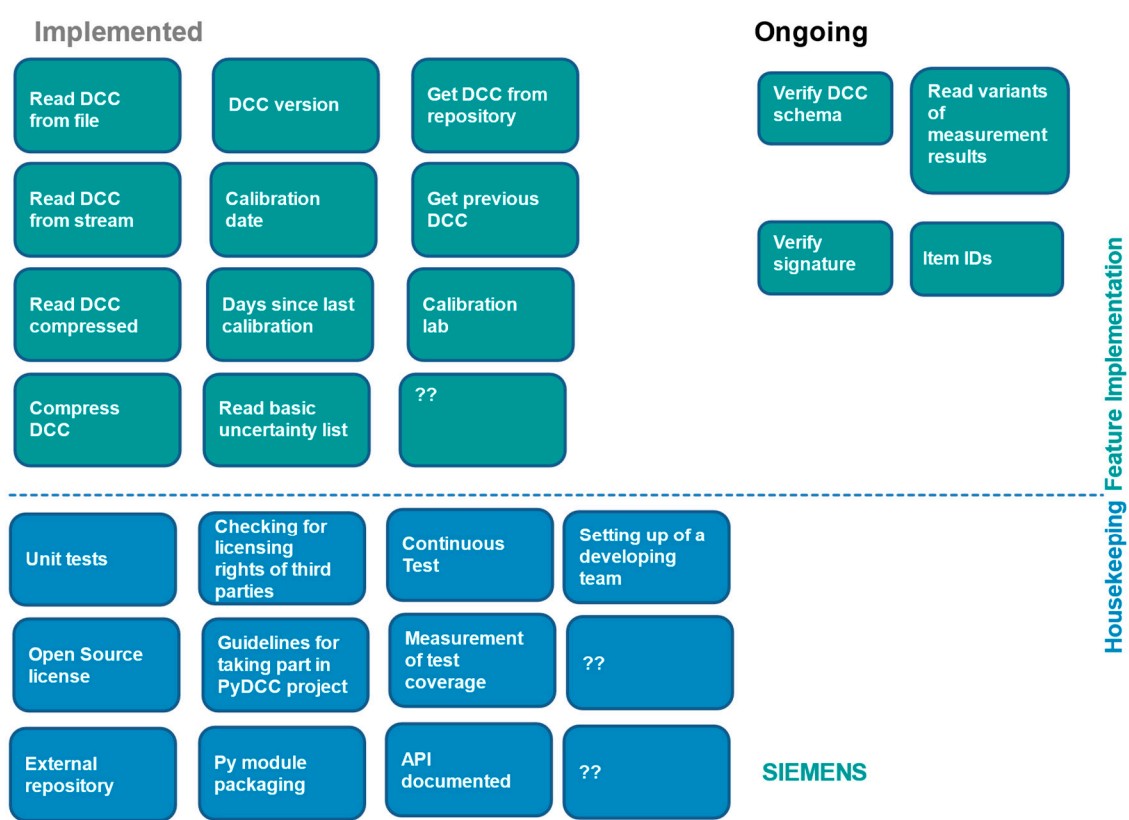

**Figure 9.** PyDCC Software Management (currently unreleased, see Tobola, Andreas "Introducing PyDCC—a Python module for the DCC" [34], p. 101).

In this way, a basic functionality is also available for using the DCC on edge and smart field devices. The idea is to grant all users easy access to the information contained in the DCC by making this software functionality (which is non-competitive) available as an open-source version that includes the standardized DCC, which is provided with an unambiguous version identification and with the respective XML schema. Publishing the approach as an open-source version, which will reach many users, is supposed to minimize the effort that is needed to implement the basic functionalities in such a way that the DCC can be used by a specific company and/or can be connected with the Enterprise Resource Planning/PLM (Product Lifecycle Management).

## 6. Conclusions & Outlook

In this article, we showed that all aspects of calibration are of great economic interest. The end user often does not realize how many components have contributed to the quality assurance for a product. Establishing the use of DCCs is a task for both the industrial and trade partners and should be achieved all over the world. When the approach of the digital calibration certificate is recognized and used internationally, the DCC will play a central role. For example, language barriers which still exist today can be eliminated by means of the digital document as the contents will be standardized and can automatically be transferred to the respective language. In this way, the DCC will make an important contribution to the internationalization of measurement technology and the calibration system.

In principle, the DCC system suggested here can also be used for other applications in measurement technology. These range from legal metrology to qualification measurements that are carried out on a system in order to prove its functionality or after it has undergone maintenance work. For this purpose, only the type or level of the issuing authority must be mentioned in the DCC document. Here, the end-to-end use of a DCC would bring about great advantages for all process partners.

Currently, establishing the DCC as a component in an asset administration shell (AAS [35]) in a sub-model is envisaged.

**Author Contributions:** Conceptualization, methodology, draft preparation and visualization, S.H., S.S., L.D., T.E. and R.B.; supervision, validation, writing—reviewing and editing, L.D. and T.E. All authors have read and agreed to the published version of the manuscript.

**Funding:** The GEMIMEG II project is funded by the German Federal Ministry for Economic Affairs and Climate Action (BMWK), grant reference GEMIMEG 01 MT20001A.

**Institutional Review Board Statement:** Not applicable.

**Informed Consent Statement:** Not applicable.

**Data Availability Statement:** See 'DCC—GoodPractice · GitLab', GitLab. Available online: https://gitlab.com/ptb/dcc/dcc-goodpractice (accessed on 16 January 2023).

**Conflicts of Interest:** The authors declare no conflict of interest.

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
