# Peer review of "The Digital Calibration Certificate (DCC) for an End-to-End Digital Quality Infrastructure for Industry 4.0"

_sci, doi:10.3390/sci5010011_

Round 1

Reviewer 1 Report

This manuscript in this current form still has some problems as follows: 

1. The format of the Abstract is a little strange.

2. Please add a section introduction to Section 1.

3. In terms of literature research, it is suggested to add the description for the following related  works:

1) Symmetric implicational algorithm derived from intuitionistic fuzzy entropy. Iranian Journal of Fuzzy Systems

2) Viewpoint-based kernel fuzzy clustering with weight information granules. IEEE Transactions on Emerging Topics in Computational Intelligence

4. Page 6:  The format of the figure is strange. It needs its name.

5. Page 9:  The format of the figure is strange. It needs its name.

Reviewer 2 Report

In their article "The Digital Calibration Certificate (DCC) for an end-to-end 2 digital quality infrastructure for Industry 4.0", the authors present and depict the role of the Digital Calibration Certificate.

The proposed article is rather a white paper, project status/end report or a technical newsletter than a scientific paper.

The article lacks the format of a scientific paper (typically something close to IMRAD) and does not follow the scientific method. The state-of-the-art is poorly described as the references are merely websites or even wikipedia.

The article lacks also novelty. The same authors or the same organization (PTB) have published similar papers with similar content on DCC:s.

Hackel, Siegfried, et al. "The digital calibration certificate." PTB-Mitteilungen 127.4 (2017): 75-81.

Eichstädt, Sascha. "PTB Digitalization Strategy." Metrologie für die Digitalisierung von Wirtschaft und Gesellschaft (2017): 41.

Hackel, Siegfried, et al. "The fundamental architecture of the DCC." Measurement: Sensors 18 (2021): 100354.

Brown, Clifford, et al. "Infrastructure for Digital Calibration Certificates." 2020 IEEE International Workshop on Metrology for Industry 4.0 & IoT. IEEE, 2020.

Reviewer 3 Report

The paper is good and very important in the current digitalization of industry and metrology. Only two very minor changes are needed.

captions of Fig 3 and 6 do not appear in the correct position in the text.

In line 274 i would change the word "realize" by "express". "Realize" has an especial meaning in this context.

Reviewer 4 Report

 DCCs are getting popular in metrology system because of the conceptual advantages, like reducing transmission errors and improving trace efficiency for manufacture. The manuscript introduces the idea and future usage of DCC which is useful for the people who work in the field of metrology.

suggestion: The word "Maschine" should be changed into English version in Fig.6&7.

Round 2

Reviewer 1 Report

It can be accepted now.

Reviewer 2 Report

No improvement to the article was made.